# Estimating smoking prevalence in general practice using data from the Quality and Outcomes Framework (QOF)

Kate Honeyford,[1] Richard Baker,[1] M John G Bankart,[2] David R Jones[1]

► Prepublication history and additional material is available. To view please visit the journal (http://dx.doi.org/10.1136/bmjopen-2014-005217).

[1]Department of Health Sciences, University of Leicester, Leicester, UK
[2]Institute of Primary Care and Health Sciences, Keele University, Keele, UK

**Correspondence to**
Kate Honeyford;
ceh28@le.ac.uk

## ABSTRACT

**Objectives:** To determine to what extent underlying data published as part of Quality and Outcomes Framework (QOF) can be used to estimate smoking prevalence within practice populations and local areas and to explore the usefulness of these estimates.

**Design:** Cross-sectional, observational study of QOF smoking data. Smoking prevalence in general practice populations and among patients with chronic conditions was estimated by simple manipulation of QOF indicator data. Agreement between estimates from the integrated household survey (IHS) and aggregated QOF-based estimates was calculated. The impact of including smoking estimates in negative binomial regression models of counts of premature coronary heart disease (CHD) deaths was assessed.

**Setting:** Primary care in the East Midlands.

**Participants:** All general practices in the area of study were eligible for inclusion (230). 14 practices were excluded due to incomplete QOF data for the period of study (2006/2007–2012/2013). One practice was excluded as it served a restricted practice list.

**Measurements:** Estimates of smoking prevalence in general practice populations and among patients with chronic conditions.

**Results:** Median smoking prevalence in the practice populations for 2012/2013 was 19.2% (range 5.8–43.0%). There was good agreement (mean difference: 0.39%; 95% limits of agreement (−3.77, 4.55)) between IHS estimates for local authority districts and aggregated QOF register estimates. Smoking prevalence estimates in those with chronic conditions were lower than for the general population (mean difference −3.05%), but strongly correlated ($R_p$=0.74, p<0.0001). An important positive association between premature CHD mortality and smoking prevalence was shown when smoking prevalence was added to other population and service characteristics.

**Conclusions:** Published QOF data allow useful estimation of smoking prevalence within practice populations and in those with chronic conditions; the latter estimates may sometimes be useful in place of the former. It may also provide useful estimates of smoking prevalence in local areas by aggregating practice based data.

## Strengths and limitations of this study

- This paper clearly demonstrates that useful estimations of smoking prevalence within practice populations can be calculated from routine data published through the Quality and Outcomes Framework (QOF).
- Our analysis shows that estimates of smoking prevalence in those with chronic conditions can be used in some situations in place of an estimate for the general population, if this is not available.
- Comparisons with local area estimates suggest QOF-based estimates are useful for estimating smoking prevalence in both practice populations and in local areas.
- QOF data rely on self-reported smoking status, recorded in the previous 27 months, which may underestimate smoking status or the effectiveness of interventions.
- This study does not have access to individual patient data limiting our understanding of patients who do not have smoking status recorded and the possible impact of missing data on estimates of smoking prevalence.

## BACKGROUND

Despite smoking prevalence in England falling 'below 20% for the first time in 80 years',[1] reducing smoking remains a key public health priority in England as in many countries, with local authorities and primary care services being expected to play a key role in local tobacco control services.[2] In addition, clinical commissioning groups (CCGs), membership organisations responsible for planning, organising and purchasing nationally funded healthcare within their local areas, have their own health targets. Reducing smoking prevalence is a key component of many targets, for example, improving chronic obstructive pulmonary disease (COPD) outcomes[3] and reducing inequalities in coronary heart disease (CHD).[4] Reliable estimates of smoking prevalence for practice populations and local

areas are useful to assess need, inform targeting of interventions delivered through primary care and to evaluate those interventions. For practices and CCGs it is important to be able to evaluate different approaches to smoking cessation and to understand the different levels of risk in different practices. Practice based estimates are of particular importance for research into a variety of health outcomes and their associations with primary care. Research of this type generally aims to take characteristics of the practice populations into account and the inclusion of smoking prevalence has been shown to be important in the interpretation of other factors, in particular socio-economic deprivation.[5][6] Currently a variety of measures of smoking prevalence in practice populations are being used[6][7] and some studies do not include a measure of smoking,[8–10] despite the recognised associations between smoking prevalence and a range of chronic conditions.[11]

## National surveys

In England, there are various national surveys of smoking prevalence including the Health Survey for England[12]; the General Lifestyle Survey[13]; the Smoking Toolkit Study (STS) and the Integrated Household Survey (IHS). The IHS began in 2009 and is a composite survey including questions on smoking habits (involving over 420 000 adults in 2011). IHS statistics are designated as experimental, in a 'testing phase' and not yet fully developed,[14] but estimates are available for local authorities. It could, therefore, be argued that there is no gold standard measure of smoking in local areas, and there are no surveys which aim to establish the smoking prevalence within practice populations.

## Patient records

Analyses of individual patient records, using the THIN (The Health Improvement Network)[15] and QRESEARCH[16] databases, provide strong evidence that smoking status within primary care medical records could be used to monitor national smoking patterns. There was good agreement between smoking prevalence based on medical records in the THIN database and those predicted by GHS: 22.4% compared to 21.8% respectively in men; 18.9% compared to 20.2% respectively in women.[17] Estimates of smoking prevalence based on the medical records in the QRESEARCH database have also shown good agreement with national surveys, in this case the Health Survey for England.[18]

## Quality and Outcomes Framework

The national Quality and Outcomes Framework (QOF) is a payment for performance system which was introduced in England in 2004 to improve the quality of primary care for patients. Practices are awarded points for achieving targets and these points are translated into financial reward. Since its inception QOF has included indicators relating to smoking.[19] The underlying aim of these indicators has not changed over the years; (a) practices should record smoking status in patient notes and (b) for those who smoke, smoking cessation advice/support/treatment should have been offered. Until 2012/2013 the focus was on targeting smoking cessation advice to those with chronic conditions. Table 1 summarises QOF smoking indicators 2006/2007–2014/2015.

The QOF indicators have not been designed to determine smoking prevalence within the practice population; indeed it is clearly stated that "QOF provides no information on numbers of smokers and non-smokers",[20] attributing this mainly to the condition-specific nature of the indicator. The wording has not changed since the inclusion of the two new indicators which apply to the general population and are not condition-specific.

## Objective

In this paper we aim to explore to what extent underlying data published as part of QOF can be used to estimate smoking prevalence within practice populations. The usefulness of these estimates are explored by (i) comparing aggregated data with local area estimates from other sources and (ii) including practice level estimates in a model of CHD mortality.

## METHOD
### Sample

All practices within three primary care trusts (PCTs), the organisational unit for administering general practices in England (2006/2007 to 2011/2012) in the East Midlands, were eligible for inclusion in the study. 215 practices with QOF data available for the 7 financial years were included in the analysis. 14 practices were excluded because they lacked data for all 7 years. One practice was excluded from the study as it served a restricted practice list; another practice was excluded from the study as 2012/2013 QOF data strongly suggested an error.

### Manipulation of QOF data

QOF data can be downloaded from the Health and Social Care Information Centre website containing information for all practices in a region[21]; table 2 illustrates the type of data available.

Two key QOF indicators are used in the calculations of smoking prevalence in the total practice population:
- SM07 'The percentage of patients aged 15 years and over whose notes record smoking status in the preceding 27 months'
- SM08 'The percentage of patients aged 15 years and over who are recorded as current smokers who have a record of an offer of support and treatment within the preceding 27 months'

**Table 1** Summary of smoking indicators for which underlying achievement is published

| General form of the indicator | Patient group<br>Patients with any, or any combination of the following conditions: CHD, stroke or TIA, hypertension, diabetes, COPD or asthma* | All patients aged 15 years+ |
|---|---|---|
| Percentage of patients whose notes record smoking status† | SM01: 2006/2007 and 2007/2008<br>SM03: 2008/2009–2011/2012<br>SM05: 2012/2013<br>SMOK002: 2013/2014–2014/2015 | Records 22: 2006/2007 and 2007/2008<br>Records 23: 2008/2009–2011/2012<br>SM07: 2012/2013<br>SMOK001: 2013/2014—retired in 2014/2015 |
| Percentage of patients who are recorded as current smokers whose notes contain a record that smoking cessation advice or referral to a specialist service, where available, has been offered within the previous 15 months‡ | SM02: 2006/2007 and 2007/2008<br>SM04: 2008/2009–2011/2012<br>SM06: 2012/2013<br>SMOK005: 2013/2014–2014/2015 | SM08: 2012/2013<br>SMOK004: 2013/2014–2014/2015 |
| The practice supports smokers in stopping smoking by a strategy which includes providing literature and offering appropriate therapy. | | Information 5: 2006/2007–2011/2012<br>SMOK003: 2012/2013–2014/2015 |

*In 2008/2009 chronic kidney disease, asthma, schizophrenia, bipolar affective disorder or other psychoses were added to the list of chronic conditions and in 2012/2013 peripheral arterial disease was added.
†For those with chronic conditions, the record must have been made in the past 15 months, reduced to 12 months in 2013/2014, for all patients the period is 27 months, reduced to 24 months in 2013/2014.
‡In 2012/2013 this changed to 'who have a record of an offer of support and treatment within the preceding 15 months', the period is 27 months for all patients, reduced to 12 months and 24 months respectively in 2013/2014.
COPD, chronic obstructive pulmonary disease; TIA, transient ischemic attack.

**Table 2** Example of QOF data from 2012/2013, showing how it can be used to calculate smoking prevalence for individual practices

| QOF description | Interpretation for purposes of calculating smoking prevalence | Example practices | | | | |
|---|---|---|---|---|---|---|
| | | A | B | C | D | E |
| SM07 points | | 11 | 10.5 | 10.8 | 9.6 | 11 |
| SM07 numerator | Patients* whose notes contain a record of smoking status | 3450 | 1319 | 6276 | 31 948 | 6504 |
| SM07 denominator | Patients who are eligible to be included in this indicator† | 3721 | 1497 | 7033 | 37 654 | 7212 |
| SM07 UA | | 92.70% | 88.10% | 89.20% | 84.80% | 90.20% |
| SM08 points | | 12 | 9.9 | 12 | 8.9 | 12 |
| SM08 numerator | Patients who are recorded as current smokers and have a record of an offer of support, etc | 1024 | 325 | 1578 | 8439 | 2165 |
| SM08 denominator | Patients who are recorded as current smokers | 1129 | 401 | 1586 | 10 931 | 2373 |
| SM08 UA | | 90.70% | 81.00% | 99.50% | 77.20% | 91.20% |
| | Calculation to determine percentage who are smokers SM08 den/SM07 den | 1129/3721 | 401/1497 | 1586/7033 | 10 931/37 654 | 2373/7212 |
| | Estimate of smoking prevalence | 30.30% | 26.80% | 22.60% | 29.00% | 32.90% |

*Patients aged over 15.
†For example, patients who are newly registered with the practices (less than 3 months) are excluded from the indicator.
QOF, Quality and Outcomes Framework.

These can be summarised as follows:

SMOKING STATUS INDICATOR (SM07) =

$$\frac{\text{No. of patients who have their smoking status recorded}}{\text{No. of eligible patients in the practice}}$$

SMOKING CESSATION INDICATOR (SM08) =

$$\frac{\text{No. of patients who have a record of cessation support}}{\text{No. of patients recorded as current smokers}}$$

The denominator of the SMOKING STATUS INDICATOR (SM07) provides an estimate of the sample of the practice population whose smoking status should be recorded. This includes the whole practice population aged over 15, with the exception of people who have joined the practice in the 3 months prior to the data extraction point and patients who refuse to provide their smoking status.

The denominator of the SMOKING CESSATION INDICATOR (SM08) provides an estimate of those who are recorded as current smokers.

In addition, indicators of a similar nature were included but applying to those with any, or any combination, of a range of QOF specified chronic conditions (SM05 and SM06).

Using the data given for these indicators, it is possible to estimate the smoking prevalence in a practice population, as summarised below.

SMOKING PREVALENCE ESTIMATE

$$= \frac{\text{No. of patients recorded as current smokers}}{\text{No. of eligible patients in the practice}}$$

$$= \frac{\text{Denominator of SM08}}{\text{Denominator of SM07}}$$

For example, for practice A the denominator for SM07 is 3721—the number of eligible patients in the practice. The denominator for SM08 is 1129, indicating that there are 1129 registered patients recorded as current smokers. Hence smoking prevalence can be estimated as 1129/3721 or 30.3%. Table 2 gives worked examples for five practices. This method was used to estimate smoking prevalence for the total practice population in 2012/13 and, using appropriate indicators, for those with chronic conditions from 2006/2007 to 2012/13 (SM05 and SM06 in 2012/13).

In addition, the percentage of the practice population with a chronic condition was determined using the denominator of SM07 as a measure of the practice population and the denominator of SM05 as a measure of the practice population with a chronic condition.

**Comparisons with local area estimates**

Practice postcodes were linked to local authority districts using the National Statistics Postcode Directory (NSPD)[22] and then confirmed by visual check of addresses. Practice level data were aggregated to estimate smoking prevalence in local authority districts. Details of the estimated population of each district, the aggregated population for which smoking status has been determined, the number of practices in each district and the sample size for the IHS 2011/2012 are included in table 3. These estimates were compared to estimates of smoking prevalence in local authority districts based on data from the IHS.[24]

**Modelling**

To determine the importance of being able to estimate smoking prevalence in practice populations, the estimate of smoking prevalence was included in a model to determine the associations of premature CHD (under 75) mortality with various population and service characteristics; the methods are described by Honeyford et al.[10] Here, counts of premature CHD deaths (between April 2006 and March 2009) were modelled using negative binomial regression, using the same explanatory variables but including estimated smoking prevalence for those with chronic conditions based on QOF 2006/2007. Service and population characteristics derived from QOF registers from 2006/2007 were originally selected for inclusion in the study but an estimate of smoking prevalence for the general population was not available for this year.

**RESULTS**
**Estimates using QOF data**
**Estimation of overall smoking prevalence using QOF smoking indicators 2012/2013**
The median underlying achievement for the recording of smoking status in the total practice population was 88.1% in 2012/2013 (IQR: (83.7, 91.0)). The median estimate of smoking prevalence in practice populations was 19.2%, ranging from 5.8% to 43.0% (IQR: (15.1%, 22.9%)).

**Estimates of smoking prevalence in those with chronic conditions using QOF smoking indicators 2012/2013**
The underlying achievement for recording smoking status in those with chronic conditions was higher than for the total practices population (96.6% IQR (95.0, 97.7)). The median practice based estimate for those with any or any combination of a specific list of chronic conditions was 15.4% (IQR: 12.6% to 19.4%), ranging from 7.1% to 51.5%.

The estimates of smoking prevalence in those with chronic conditions have been consistent since 2006/2007, with the median varying slightly during that time. Concordance was high between estimates for all years; Lin's concordance coefficient[25] was greater than 0.92 and mean difference was less than1 in all cases (see online supplementary table S1 in appendix for more details).

**Comparisons with local area estimates**
Estimates of smoking prevalence were in line with estimates derived from the IHS. Aggregating over the total

**Table 3** Comparison of the population of each district based on the 2011 Census and aggregation QOF based practice data

| Local authority | Population aged 15 and over (2011 Census)* | Population included in QOF indicator SM07† | Number of general practices‡ | IHS sample size 2011/2012§ |
|---|---|---|---|---|
| Leicestershire | | | | |
| Blaby | 77 600 | 67 895 | 9 | 301 |
| Charnwood | 139 800 | 152 533 | 24 | 396 |
| Harborough | 70 200 | 69 168 | 8 | 234 |
| Hinckley and Bosworth | 87 800 | 84 159 | 12 | 305 |
| Melton | 41 900 | 34 912 | 2 | 130 |
| North West Leicestershire | 77 000 | 78 331 | 14 | 242 |
| Oadby and Wigston | 47 100 | 48 054 | 9 | 167 |
| Northamptonshire | | | | |
| Corby | 49 400 | 57 112 | 5 | 131 |
| Daventry | 64 100 | 71 902 | 8 | 223 |
| East Northamptonshire | 70 900 | 55 279 | 8 | 217 |
| Kettering | 75 900 | 87 059 | 9 | 180 |
| Northampton | 171 600 | 184 370 | 27 | 446 |
| South Northamptonshire | 69 700 | 60 391 | 8 | 205 |
| Wellingborough | 61 300 | 61 013 | 9 | 172 |
| Unitary Authorities | | | | |
| Leicester | 264 600 | 293 156 | 59 | 1475 |
| Rutland | 31 300 | 29 628 | 4 | 416 |
| Totals | 1 400 200 | 1 434 962 | 215 | 5240 |

*Data based on 2011 Census available from ONS.[23]
†Based on QOF registers accessed from ref. [21]
‡Practices are matched to local authority districts based on the postcode of the practice.[22]
§Based on IHS data 2011/2012.[24]
QOF, Quality and Outcomes Framework; IHS, integrated household survey.

area, smoking prevalence was 19.5%, compared to 19.3% when IHS district level data were aggregated over the same area. When practice data were combined to give estimates of smoking for local authority districts there was a strong positive correlation ($R_p$=0.86, p<0.0001) and good agreement (mean difference: 0.39%; 95% limits of agreement (−3.77, 4.55)) between estimates based on QOF registers and IHS estimates (figure 1).[26]

When the estimates of prevalence for those with chronic conditions were aggregated into local authority districts, estimates were lower than IHS estimates for the majority of areas.

### Associations between measures
Association between smoking prevalence in the general practice population and those with chronic conditions
Smoking prevalence in those with chronic conditions was lower than in the general practice population. The mean difference between the two estimates was −3.05% (95% limits of agreement: (−8.65, 1.56)).The Bland-Altman plot does not suggest a strong pattern, despite some evidence that the difference increases as the average increases (figure 2). There was a strong positive correlation ($R_p$=0.92, p<0.0001) between the overall estimate of smoking prevalence within a practice population and in those with chronic conditions. A regression

model was developed to predict smoking prevalence in the general population based on the prevalence in those with chronic conditions; removal of outliers improved model fit.

### Associations between recording of smoking status and prevalence
There was a strong positive correlation between recording of smoking status in the general population and in those with chronic conditions (underlying achievement for SM07 and SM05 respectively) ($R_p$=0.74, p<0.0001). There was no evidence of an association between smoking prevalence in the general population and recording of smoking status ($R_p$=−0.07, p=0.28) or the percentage with a chronic condition ($R_p$=0.03, p=0.67).

### Including smoking prevalence estimates in models of mortality
Table 4 shows incident rate ratios (IRRs), 95% CIs and associated p values for the original and modified models. Inclusion of the smoking prevalence variable in the model reduced the strength of the associations between deprivation and premature mortality, and percentage white and premature mortality. Sensitivity analysis considering the impact of exception reporting indicates no impact on interpretation (see Doran et al[27] for details of exception reporting).

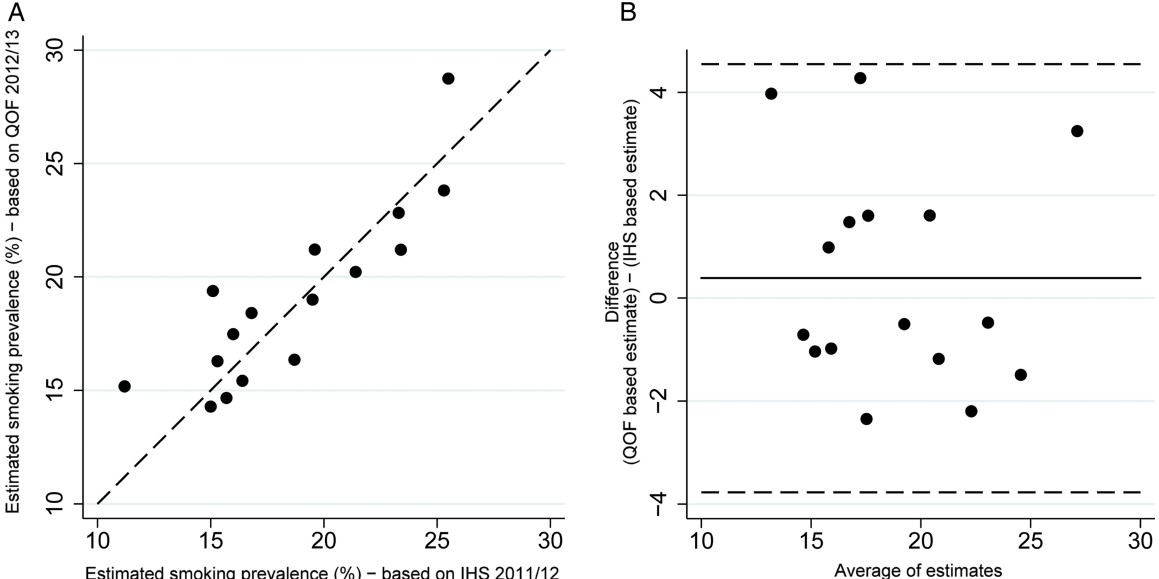

**Figure 1** Relationship between aggregated QOF estimates and IHS estimates for local authority districts. (A) Association between estimates (dashed line: estimates are equal). (B) Bland-Altman plot showing relationship between difference in estimates and mean difference (solid line: mean difference; dashed lines: 95% limits of agreement). QOF estimates based on 2012/2013 data; IHS estimates based on 2011/2012 survey. QOF, QOF, Quality and Outcomes Framework; IHS, integrated household survey

## DISCUSSION
### Principal findings
These results show how the QOF registers required as part of the general practice pay for performance scheme in England can be used to estimate smoking prevalence in practice populations and that these estimates are useful when analysing patterns of mortality. Practice based estimates can be aggregated to provide estimates of smoking prevalence in local areas.

When smoking prevalence is estimated in the general population using QOF indicators there is good agreement with estimates of IHS smoking prevalence for similar geographical areas.

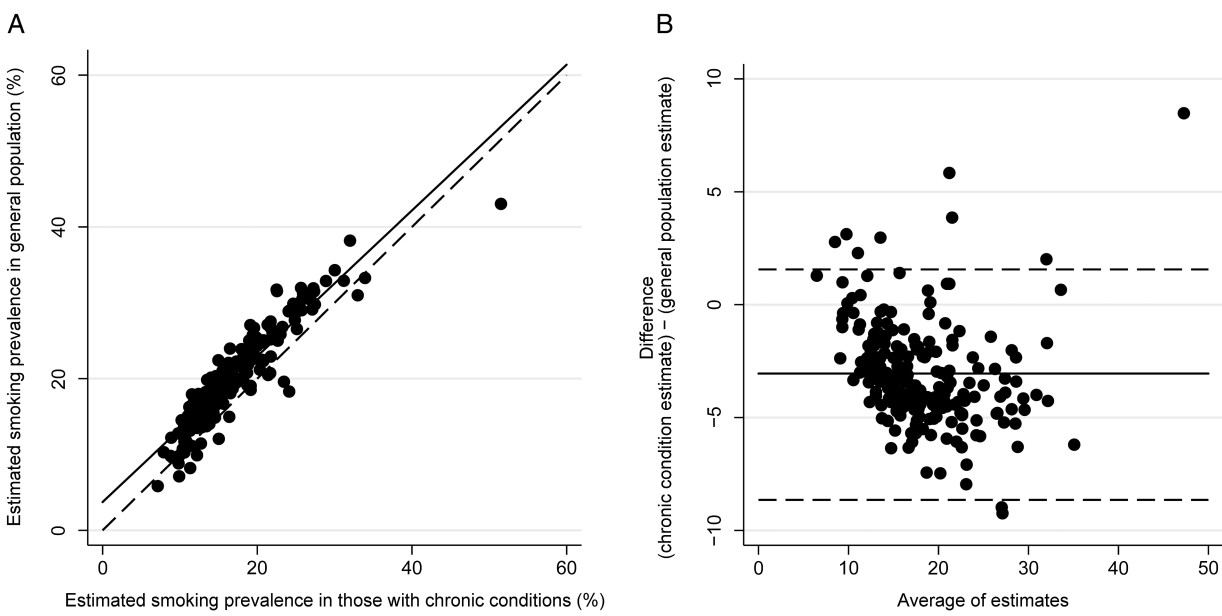

**Figure 2** Relationship between QOF estimates for the general population and those with chronic conditions (2012/2013), (A) Association between estimates (dashed line: estimates are equal; solid line: fitted line). (B) Bland-Altman plot showing relationship between difference in estimates and mean difference (solid line: mean difference; dashed lines: 95% limits of agreement). SM07 and SM08 (2012/2013) used for QOF estimates for the general population; SM05 and SM06 (2012/2013) used for QOF estimates for those with chronic conditions. QOF, QOF, Quality and Outcomes Framework

**Table 4** Estimated incident rate ratios (IRRs) for premature (U75) CHD mortality count (n=215)*

| Explanatory variable | Without smoking prevalence variable | | | With smoking prevalence variable | | |
|---|---|---|---|---|---|---|
| | IRR | 95% CI | p Value | IRR | 95% CI | p Value |
| Percentage white patients | 1.007 | (1.003 to 1.012) | 0.002 | 1.001 | (0.995 to 1.007) | 0.657 |
| Deprivation score (IMD 2007) | 1.017 | (1.011 to 1.024) | <0.0001 | 1.005 | (0.995 to 1.015) | 0.348 |
| Prevalence of diabetes (QOF 2006/2007) | 1.108 | (1.020 to 1.203) | 0.015 | 1.095 | (1.008 to 1.187) | 0.031 |
| Percentage over 65 | 1.060 | (1.038 to 1.083) | <0.0001 | 1.067 | (1.044 to 1.091) | <0.0001 |
| Percentage male patients | 1.073 | (1.035 to 1.111) | <0.0001 | 1.058 | (1.021 to 1.097) | 0.002 |
| Number of GPs per 1000 patients | 1.209 | (0.894 to 1.637) | 0.218 | 1.113 | (0.821 to 1.508) | 0.491 |
| Hypertension detection 2006/2007 (QOF 2006/2007) | 0.984 | (0.955 to 1.014) | 0.300 | 0.988 | (0.959 to 1.018) | 0.416 |
| Percentage of patients offered smoking cessation advice (SM02—QOF 2006/2007) | 1.006 | (0.996 to 1.016) | 0.271 | 1.010 | (1.000 to 1.021) | 0.057 |
| Percentage of serum cholesterol (CHD08—QOF 2006/2007) | 0.989 | (0.980 to 0.999) | 0.028 | 0.992 | (0.983 to 1.002) | 0.109 |
| Percentage of aspirin (CHD09—QOF 2006/2007) | 1.007 | (0.986 to 1.029) | 0.514 | 1.003 | (0.982 to 1.025) | 0.777 |
| Percentage of patients with recalled perception of being able to see preferred GP (QOF 2006/2007) | 0.995 | (0.990 to 1.000) | 0.069 | 0.995 | (0.990 to 1.000) | 0.061 |
| Percentage of smoking prevalence—estimated (QOF 2006/2007) | | | | 1.031 | (1.012 to 1.052) | 0.002 |

*IRR, 95% CIs and associated p values as a result of negative binomial model of count of premature mortality caused by CHD.
CHD, coronary heart disease; GP, general practitioner; QOF, Quality and Outcomes Framework.

QOF data can also be used to estimate smoking prevalence in those with chronic conditions, which is generally lower than smoking prevalence in the general population. There is good agreement between the estimates in successive years. The correlation between estimates of smoking prevalence in the general population in 2012/2013 and those with chronic conditions is strong. These strong correlations suggest that the estimates based on previous years can be used in place of smoking prevalence in the general population for some purposes. Regression analysis suggests that smoking prevalence in those with chronic conditions can be used to predict smoking prevalence in the general practice population, for practices with a typical patient list.

When an estimate of smoking prevalence in those with chronic conditions was used in a study of the association between premature CHD mortality and various population and service characteristics an important positive association between CHD mortality and smoking prevalence was shown.

## Strengths and weaknesses

The agreement between IHS based area estimates of smoking prevalence and estimates based on combining QOF data provides evidence to suggest that manipulating QOF data results is a useful measure of smoking prevalence within practice populations when compared to other available measures. This is supported by the work of Szatkowski et al[17] which found good agreement between national smoking prevalence predicted by patient records and the General Household Survey. In addition, practice based QOF data can be aggregated to

provide local area estimates of smoking prevalence based on a much larger sample size than other surveys.

When comparing practices and analysing patterns across practices, it is important that the estimate is consistent across practices. The percentage of patients who do not have their smoking status recorded varies from 40% to less than 1%, but the characteristics of these patients are not known. Recording of smoking status[18 19 28] has been shown to vary between groups; women, older people and those with chronic conditions were more likely to have their smoking status recorded. National surveys suggest that smoking rates are lower in these groups and therefore smoking prevalence from QOF may underestimate actual smoking prevalence. The implications of this will vary between practices, dependent on the proportion of these groups within their practice populations. Our analysis did not find an association between the percentage with a chronic condition and the recording of smoking status in the total population or the estimate of smoking prevalence.

QOF data are based on self-reported smoking status, which has been shown to be reliable in the general population,[28] but to underestimate smoking prevalence in pregnant women.[29] In addition, practices are only asked to record smoking status in the preceding 27 months, meaning the estimates may be useful in assessing need and analysing associations, but will have disadvantages in assessing the effectiveness of interventions, unless practices commit to more regular recording.

Practice level smoking data have been aggregated to local authority districts based on practice postcode rather than patient postcodes. General practice

catchments are not constrained by local authority boundaries; however, studies have shown that 80% of patients live within a 10 min car journey of their practice,[30] suggesting that patients choose practices close to where they live. It is relatively common for practice postcodes to be used as a proxy for patient postcodes; however, when used to estimate deprivation this has been found to underestimate relationships between deprivation and health outcomes.[31 32]

Further work using individual patient records is necessary to analyse the frequency of recording of smoking status and the characteristics of patients for whom no smoking status is recorded or have been excluded on the basis of exception reporting. In this analysis practice level data have been aggregated to estimate smoking prevalence in local authority districts. Analysis of patient level postcode information, not available for this study, would allow estimates of smoking prevalence for smaller geographical areas to be made. These could then be compared to modelled estimates or locally commissioned surveys, where they exist.

## Implications

Manipulating QOF data is an easy and cost-effective method of estimating smoking prevalence in both practice populations and local areas, although further work is necessary to determine the validity of using aggregated practice level data for local area estimation. Both local area and practice based estimates are important to those tasked with reducing smoking rates and improving the nation's health. CCGs and public health departments in local authorities need them to target smoking cessation and other additional resources. Understanding more about the patient populations would enable similar practices to be compared when considering differences in health outcomes and the apparent effectiveness of interventions.[33]

Current estimates of smoking prevalence in local areas are based on the Integrated Household Study. The IHS is currently in an experimental phase since the weighting methodology needs to be assessed and potentially revised.[34] Aggregated practice level data include the majority of the resident adult population in local areas and could therefore be a more useful measure of local area smoking prevalence, at district level and at smaller local areas than are currently available through the IHS. Analysis of patient level geographical data is necessary to determine the potential utility of simple and more complex aggregation methods.

When estimates of smoking prevalence are included in the analysis of the associations between premature CHD mortality and practice population and service characteristics, there are reductions in the magnitude of the IRRs for both deprivation and percentage white. This suggests that these may be acting as surrogate markers of other lifestyle factors, such as smoking prevalence. Hence, the lack of reliable smoking information may be leading to relative over emphasis being placed

on socio-economic deprivation, often described using an index of multiple factors. Reliable measures of smoking prevalence will improve our understanding of the relative importance of deprivation and other characteristics in explaining inequalities in a variety of health outcomes.

Smoking prevalence in those with chronic conditions is typically lower than in the general population. This may be due to diagnosis increasing motivation to quit smoking,[35] the increase in smoking cessation advice and support[36] or the age and gender profile of those with chronic conditions. Smoking prevalence in those with chronic conditions has not reduced over the 7 year period covered in this analysis, possibly suggesting that smoking cessation advice has limited effect, but this may be due to the turnover of patients with chronic conditions as a result of both premature mortality and new diagnoses. A wide range of smoking cessation advice and support has recently been reviewed by Zwar et al[37]; consideration of how these impact on those with chronic conditions is recommended as a result of this finding.

QOF smoking indicators have changed since 2004 and continue to change. The introduction, in 2012/2013, of an indicator which allows estimates of the smoking prevalence within the general population is useful for researchers as well as CCGs and public health officials. The removal of the indicator that covers the recording of smoking status in the total population from QOF in 2014/2015 will impact on the methodology described in this paper, although the number of patients who are recorded as current smokers will continue to be available. The population of the practice will need to be used as the denominator in the calculation of smoking prevalence. It will be important to determine if the smoking status declines after the removal of the indicator; a recent study suggests that removal of indicators does not lead to a decline in clinical activities.[38]

## CONCLUSION

Data published through QOF allow useful estimations of smoking prevalence within practice populations and in those with chronic conditions to be made. These estimates are important in developing our understanding of differences in health outcomes between practices, and are useful to both individual practices and CCGs when comparing practice level health outcomes, to assess need and to inform targeting. Aggregating practice level data may also be useful to allow estimates of smoking prevalence in local areas to be made. Revisions to QOF means that researchers will need to update methodology as indicators change.

**Contributors** The study was conceived by KH, RB, MJB and DRJ. KH designed the study, carried out the analysis and drafted the initial manuscript. MJB and DRJ contributed to the statistical analysis. RB, MJB and DRJ contributed to drafting and editing the final manuscript and interpreting and reviewing the results of the analysis.

**Funding** The research was funded and led by National Institute for Health Research (NIHR) Collaboration for Leadership in Applied Health Research and Care) based at LNR.

**Disclaimer** The views expressed are those of the authors and not necessarily those of the NHS, the NIHR or the Department of Health. The funders had no role in the study design, data collection and analysis, decision to publish or preparation of the manuscript.

**Competing interests** KH had financial support from CLAHRC in the form of funding for PhD fees. RB is in receipt of an NIHR Senior Investigator award.

**Ethics approval** NRES advised that NHS ethics committee was not required.

**Provenance and peer review** Not commissioned; externally peer reviewed.

**Data sharing statement** No additional data are available.

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
