## [Reviewer comments · BMJ Open]

Some articles will have been accepted based in part or entirely on reviews undertaken for other BMJ Group journals. These will be reproduced where possible.

ARTICLE DETAILS

TITLE (PROVISIONAL)	Estimating smoking prevalence in general practice using data from the Quality and Outcomes Framework (QOF)
AUTHORS	Honeyford, Kate; Baker, Richard; Bankart, M. John; Jones, David

VERSION 1 - REVIEW

REVIEWER	Tim Coleman University of Nottingham
REVIEW RETURNED	04-Apr-2014

GENERAL COMMENTS	This is potentially a useful paper as it illustrates how routinely collected primary care data which is freely available via QOF reporting can be used to estimate smoking prevalence in small areas (using data aggregated from a number of practice returns). This is cheaper than commissioning a local survey of smoking prevalence and doesn't require access to, or manipulation of, a large routine dataset of GP records (e.g. THIN). As such the paper reports a method for estimating locality smoking prevalence that could be widely and easily used. Unfortunately the methods for aggregating data aren't described well and some key components of the method require elaboration (e.g. how many practices' data were aggregated into units for comparison?). Additionally, the paper provides a lot of findings from a modelling exercise which, I think, is aimed at demonstrating the utility of the prevalence data. However, I think the manuscript would be improved by giving more details regarding methods of prevalence determination and focussing less on the modelling aspects of the research. Some specific comments follow but, overall, I think this paper has potential utility and would encourage revision and re-submission. The title doesn't really describe the paper adequately. The prevalence of smoking in general practices isn't being estimated, rather practice data is being aggregated and being compared with locality level estimates, so GP data is being used for 'small area' estimation of prevalence. The background could make a stronger case for why estimating practice based as opposed to population based smoking rates might be useful.
---

	This sentence is illogical: “Currently a variety of measures of smoking prevalence in practice populations are being used [5-6] and some do not include smoking” I found the description of QOF data manipulation a little unclear and in places the word ‘denominator’ rather than ‘numerator’ is used. The following sentence was a little confusing: “In this analysis the denominator of the smoking cessation indicator was used as a measure of the number of people who smoke; the population was based on the denominator of the smoking status indicator”. I couldn’t really understand why the ‘modelling’ (P5) was undertaken and this aspect of the paper was neither justified in the introduction or nor anticipated within stated objectives. Despite this, a large proportion of the paper’s results are given to reporting findings from this. Conversely, findings from the comparison of practice based, aggregated data measures of prevalence with those from comparable small areas, which is the only question justified in the introduction, are described in seven lines and one figure. The Discussion should address the fact that IHS estimates are currently ‘experimental’; there’s probably no ‘gold standard’ available for comparing small area prevalence estimates for smoking rates and this requires quite careful elaboration.
--	--

REVIEWER	Sofia Ravara Health Sciences School, University of Beira Interior Covilhã, Portugal
REVIEW RETURNED	14-May-2014

GENERAL COMMENTS	The study addresses a relevant issue assessing primary care quality and management. The methodology is sound and the paper is clearly written. I suggest minor changes to improve readability. The paper addresses specific UK primary care system measures/outcomes and often uses abbreviations. For non-UK readership it is difficult to understand them fluently. Title: I suggest to delete the abbreviation QOF and to indicate the type of study/analysis. Abstract: On the methods section line 14, I suggest defining the abbreviation PCTs. On the results section line 25, I suggest using the past tense when describing “smoking prevalence...chronic conditions are lower...” Background: I suggest including brief information explaining that in the UK GPs’ remuneration is directly linked to clinical performance targets evaluated by a set of indicators called the QOF; which justifies the completeness of these indicators. Methods: on line 52 include the citation of Honeyford et al, 2013. Results: on line 19, I suggest using the past tense when describing “smoking prevalence...chronic conditions are lower...” Conclusions; strengths and weakness: I suggest that the two first bullet statements on the strengths and weakness section would be rephrased as conclusions.
---

VERSION 1 – AUTHOR RESPONSE

Reviewer: 1

Reviewer Name Tim Coleman

1. This is potentially a useful paper as it illustrates how routinely collected primary care data which is freely available via QOF reporting can be used to estimate smoking prevalence in small areas (using data aggregated from a number of practice returns). This is cheaper than commissioning a local survey of smoking prevalence and doesn't require access to, or manipulation of, a large routine dataset of GP records (e.g. THIN). As such the paper reports a method for estimating locality smoking prevalence that could be widely and easily used.

We appreciate the suggestion that QOF data can also be used to estimate local area smoking prevalence and have aimed to include references to this in the revised draft. The main focus of our paper is to estimate smoking prevalence in practice populations rather than strictly geographically defined populations such as local areas, and we have used comparisons of aggregated practice level data with IHS estimates to validate the QOF based estimates. We therefore believe the main thrust of the paper is appropriate and the title reflects this. However, we can see that there is a potential utility in being able estimate local area estimates from QOF registers and have therefore aimed to respond to the comments to take this into account. Please see below for specific changes.

2.a Unfortunately the methods for aggregating data aren't described well and some key components of the method require elaboration (e.g. how many practices' data were aggregated into units for comparison?).

The methods section has been rewritten to clarify the steps involved, please see point 6 below for details.

2.b Additionally, the paper provides a lot of findings from a modelling exercise which, I think, is aimed at demonstrating the utility of the prevalence data. However, I think the manuscript would be improved by giving more details regarding methods of prevalence determination and focussing less on the modelling aspects of the research. Some specific comments follow but, overall, I think this paper has potential utility and would encourage revision and re-submission.

In light of the reviewer's comments we have reduced the detailed discussion of the results of the modelling exercise. We think the results and discussion which are now included, and do not take up many lines, are key to justifying the need for practice based estimates of smoking prevalence and demonstrate their importance in research. Please see below (response to paragraph 7) for specific changes

3. The title doesn't really describe the paper adequately. The prevalence of smoking in general practices isn't being estimated, rather practice data is being aggregated and being compared with locality level estimates, so GP data is being used for 'small area' estimation of prevalence. We believe the main thrust of the paper is the manipulation of QOF data to estimate smoking prevalence in general practice populations, we have improved the title to more accurately reflect the content of the paper.

Revised title: Estimating smoking prevalence in general practice using data from the Quality and Outcomes Framework (QOF)

4. The background could make a stronger case for why estimating practice based as opposed to population based smoking rates might be useful.

Paragraph 2 of Background has been strengthened to demonstrate the importance of practice based estimates to both research and CCGs and general practices.

5. This sentence is illogical: "Currently a variety of measures of smoking prevalence in practice populations are being used [5-6] and some do not include smoking"
Last sentence of paragraph 2 of background has been corrected.

6. I found the description of QOF data manipulation a little unclear and in places the word 'denominator' rather than 'numerator' is used. The following sentence was a little confusing: "In this analysis the denominator of the smoking cessation indicator was used as a measure of the number of people who smoke; the population was based on the denominator of the smoking status indicator".
The methods section has been rewritten to improve clarity.
Section 'manipulation of QOF data' in methods has been rewritten.

7. I couldn't really understand why the 'modelling' (P5) was undertaken and this aspect of the paper was neither justified in the introduction or nor anticipated within stated objectives. Despite this, a large proportion of the paper's results are given to reporting findings from this. Conversely, findings from the comparison of practice based, aggregated data measures of prevalence with those from comparable small areas, which is the only question justified in the introduction, are described in seven lines and one figure.

The main emphasis of the paper is to estimate practice based smoking prevalence and we have modified the objective to make this clearer. However, we accept the reviewers' comments that there was too much detail included regarding the results of the model. Therefore, we have reduced the detailed reporting and discussion of the results of the model; we believe the inclusion of the model demonstrates the importance of being able to estimate smoking prevalence at practice level. The objective has been modified to ensure the reader is expecting this analysis.

Objective in Introduction rewritten. Detail deleted from paragraph 6 of results (including smoking prevalence estimates in models of mortality) Detail deleted from paragraph 3 of Implications. Table 2 added.

8. The Discussion should address the fact that IHS estimates are currently 'experimental'; there's probably no 'gold standard' available for comparing small area prevalence estimates for smoking rates and this requires quite careful elaboration.

We have discussed the nature of the IHS data. However, the main emphasis of the paper is practice level estimates.

Inserted as paragraph 2 of Implications

Reviewer: 2

Reviewer Name Sofia Ravara

Institution and Country

Comments to the authors

The study addresses a relevant issue assessing primary care quality and management. The methodology is sound and the paper is clearly written. I suggest minor changes to improve readability. The paper also addresses specific UK primary care system measures/outcomes often using abbreviations. For non-UK readership it is difficult to understand them fluently.

We thank the reviewer for the comments to improve readability, and hope the changes we have made improve matters for non-UK readers.

Title: I suggest to delete the abbreviation QOF and to indicate the type of study/analysis.

We have modified the title and it indicates that the study is one making use of routinely-collected data.

We have modified the title: Estimating smoking prevalence in general practice using data from the Quality and Outcomes Framework (QOF)

Abstract:

On the methods section line 14, I suggest defining the abbreviation PCTs. On the results section line 25, I suggest using the past tense when describing “smoking prevalence...chronic conditions are lower...”

Change made

Background: I suggest including brief information explaining that in the UK GPs’ remuneration is directly linked to clinical performance targets evaluated by a set of indicators called the QOF; which justifies the completeness of these indicators.

Line added to paragraph five of Background.

Methods: on line 52 include the citation of Honeyford et al, 2013.

Change made

Results: on line 19, I suggest using the past tense when describing “smoking prevalence...chronic conditions are lower...”

Change made

Conclusions; strengths and weakness: I suggest that the two first bullet statements on the strengths and weakness section would be rephrased as conclusions.

We believe the Discussion and Conclusion sections are clearly identifiable and that the content of both Conclusions and Discussion sections are easily identifiable and appropriate.